# The Effect of Seat Layout on the Interaction of Passengers Inside the Train Carriage: An Experimental Approach for Urban Services

Sebastian Seriani [1,*], Vicente Aprigliano [1], Shirley Gonzalez [1], Gabriela Baeza [1], Ariel Lopez [2] and Taku Fujiyama [3]

1 Escuela de Ingeniería de Construcción y Transporte, Pontifica Universidad Católica de Valparaíso, Valparaíso 2362804, Chile; vicente.aprigliano@pucv.cl (V.A.); shirley.gonzalez.z@mail.pucv.cl (S.G.); gabriela.baeza.8a@gmail.com (G.B.)
2 Facultad de Arquitectura y Urbanismo, Universidad de Chile, Santiago 8330015, Chile; ariellopez@ug.uchile.cl
3 Faculty of Civil, Environmental and Geomatic Engineering, University College London, Gower St., London WC1E 6BT, UK; taku.fujiyama@ucl.ac.uk
* Correspondence: sebastian.seriani@pucv.cl

**Abstract:** The platform–train interface (PTI) is one of the most complex spaces in metro stations. At the PTI, the interaction of passengers boarding and alighting reaches high density, affecting the boarding and alighting time, among other variables related to safety and efficiency. Developing research was performed to study the effect of seat layout on the interaction of passengers by means of experiments in a controlled environment. The laboratory facility included a mock-up of a train carriage and its adjacent platform. The train was representative of urban services in the Valparaiso Metro (Chile). The results showed that seat layout changed the patterns of interaction of passengers inside the train carriage. If seats were parallel to the movement of the train, then wider corridors inside the train were generated, and therefore, the number of passengers using this space could increase up to three times. However, in urban services, passengers were located closer to the train doors to be prepared for alighting, and therefore, the passenger numbers at the central hall remained the same with the seat layout. In addition, most passengers always used seats even if they were in a different position due to the aforementioned reasons. Further research will include passengers with reduced mobility and remaining inside the train while others are alighting to identify the effect of the space used on the interaction of passengers inside the train.

**Keywords:** passenger; train; seat layout; interaction; boarding; alighting; experiments

## 1. Introduction

Boarding and alighting are still some of the most critical processes in urban railway stations (e.g., metro stations), affected mainly by the interactions between passengers at the platform–train interface (PTI) [1,2]. The manner in which passengers navigate between the platform and the train (boarding) or from the train to the platform (alighting) is a crucial concern that impacts the efficiency and safety of operations at the PTI [3,4].

In the case of the Valparaiso Metro, more than 90 thousand passengers are reached daily in the system, and most of the journeys are registered in urban areas (e.g., passengers who travel a few stations to reach their destination) [5]. Similarly, passengers in other railway systems have been reaching high levels of interaction. For instance, in the United Kingdom, there are over 4.5 million passengers boarding and alighting each year, leading to an elevated risk of passenger accidents, particularly at the PTI [6,7].

At the PTI, measures for crowd management can be introduced to minimize the interaction between boarding and alighting of passengers. Crowd management in public transport systems refers to the systematic administration of people's movement, aimed at

fostering appropriate behavior to enhance the utilization of pedestrian infrastructure [8]. The authors used sensor technologies to evaluate measures according to the space in which they are implemented. For example, at the PTI, markings on the floor can be implemented to distribute passengers waiting to board the train or to give way to those who are alighting before boarding the train (see Figure 1a). Another example is the use of priority zones for those passengers with reduced mobility (see Figure 1b).

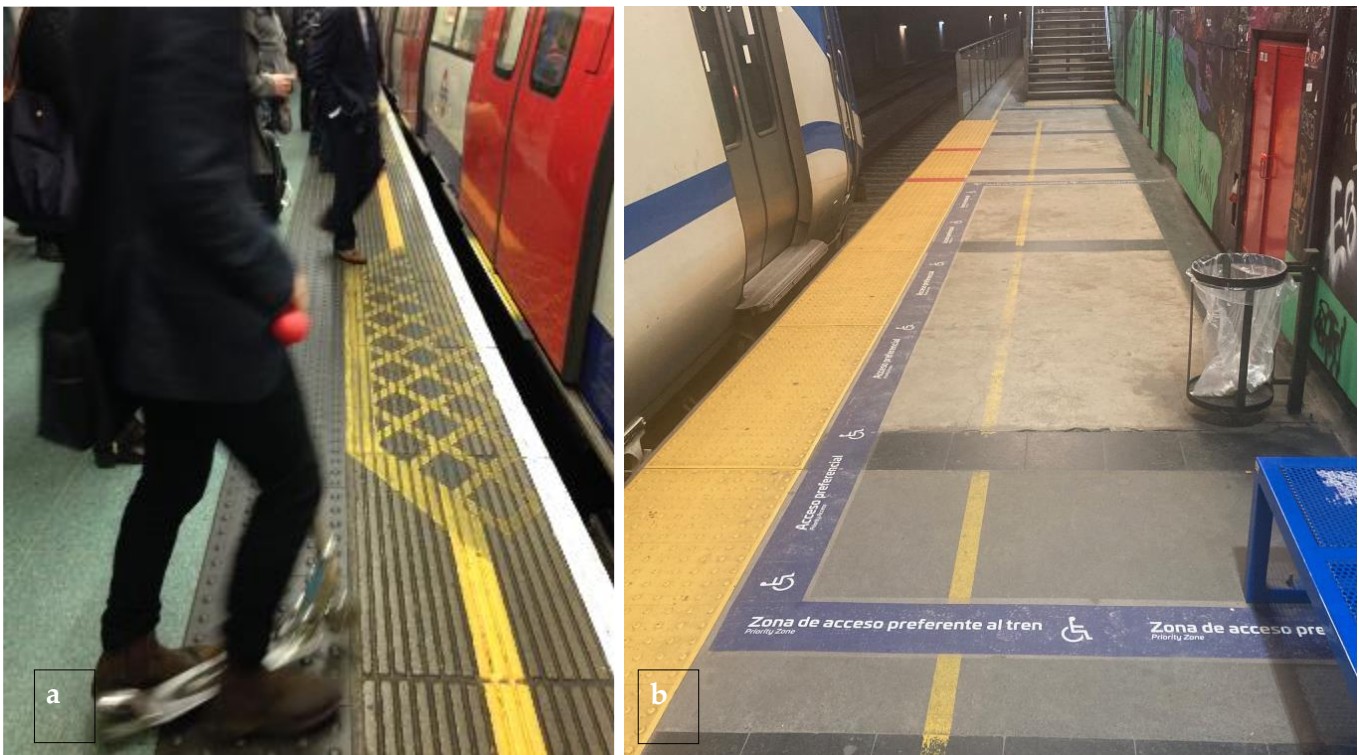

**Figure 1.** Crowd management measures: (**a**) "hatch" layout on the platform in King's Cross St. Pancras Station (London Underground) and (**b**) "priority zone" when waiting to board the train at the platform in Miramar Station (Valparaiso Metro).

While acknowledging the advantages of incorporating crowd management measures, there has been a documented effort in conducting experimental research to comprehend their impact on the interaction between passengers during boarding and alighting, considering the formation of flow lanes, distance between passengers, and their distribution on the platform [9] However, the same authors suggested exploring the interaction of passengers inside the train, which is a determinant for efficiency and safety reasons. For instance, as reported by [9], it is necessary to study the seat layout inside the train carriage, which may affect the interaction of passengers in high-density situations (e.g., around two passengers/m$^2$).

This research aimed to study the effect of seat layout on the interaction between passengers boarding and alighting inside a train carriage, considering an experimental approach focused on urban services. To accomplish this, a sample of stations were selected in the Valparaiso Metro to represent urban services within the cities of Valparaiso and Viña del Mar and identify those variables affecting the interaction of passengers inside the train carriage. Then, these variables were represented in a mock-up to represent the train carriage and its adjacent platform. Next, different experiments were performed in a laboratory environment based on scenarios of seat layouts inside the train carriage. Finally, these experiments evaluated passenger behavior using a survey focused on those volunteers who boarded and alighted the train carriage in the laboratory facility.

The paper is organized into five sections. Section 2 presents studies on the interactions between passengers during boarding and alighting at the PTI. Following that, Section 3

provides an explanation of the experimental method. Section 4 delves into the analysis of results, culminating in the conclusions presented in Section 5.

## 2. Literature Review

In assessing the interactions between passengers during boarding and alighting at the platform–train interface (PTI), the commonly employed metric is the level of service (LOS) [10]. This indicator serves as a representation of passenger congestion in various spaces, considering variables such as density, speed, and flow. The level of service (LOS), ranging from level A to level F, designates LOS E as the capacity threshold. While this metric is commonly referenced in various manuals [11,12], it is essential to recognize that overall density may not offer the most precise representation of passenger interaction at the platform–train interface. As highlighted in [9], factors such as the formation of flow lanes, the proportion of passengers boarding in relation to those alighting, passenger pressure during boarding, spacing between passengers, and their positioning on the platform and inside the train significantly influence the dynamics of passenger interaction.

When the LOS reaches a density of more than two passengers per square meter, the risk of accidents increases, affecting the safety of passengers boarding and alighting [13]. Passengers could fall onto the tracks, slip, or become stuck between the train and the platform. Therefore, different strategies are needed to improve the safety at the PTI [14]. For example, the vertical handrail inside the train should not be in front of the entrance hall, which may generate an obstacle in boarding and alighting as passengers could stay closer to the train doors. For passengers with reduced mobility, these elements should be displaced to the corridor of the train, facilitating accessibility when boarding or alighting [15].

As outlined in [16], a distinction exists between density, pertaining to the physical characteristics of the environment, and crowding, which is a psychological phenomenon. Notably, a high-density situation does not necessarily translate to a perception of crowding accompanied by elevated stress levels. The authors have put forth a model that considers both the density level and the perception of crowding and stress. Moreover, their work explores the correlation between crowding and safety. In a similar vein, [17] conducted a study on high density and stress during train commutes, specifically in situations where passengers are required to sit in close proximity. The findings revealed that passengers experienced heightened stress levels as the density increased.

To gauge crowding dynamics at the PTI, it is crucial to assess the discomfort experienced by passengers [18]. The authors utilized interviews as a tangible measure, drawing upon Fruin's level of service (LOS) from 1971 and considering the extent of crowding both on the platform and inside the train. Similarly, the authors delved into the psychological dimensions of crowds, encompassing descriptors like dense, disorderly, confining, chaotic, disturbing, cluttered, and unpleasant. The evaluation extends to the environmental conditions surrounding the crowd, including attributes like stuffiness, unpleasant odors, noise, and heat. Additionally, [19] explores how crowds respond in specific situations, using terms such as squashed, tense, uncomfortable, distracted, frustrated, restricted, hindered, stressful, and irritable.

To evaluate the interactions between passengers boarding and alighting, most of the studies reported in [1,4] are based on observations and models of existing stations, which are limited to the train layout without being able to test other configurations. To solve this problem, researchers have been experimenting in laboratory facilities where one variable is changed (e.g., markings on the platform floor) while the rest remain unchanged. One of the first types of experiments was reported by [20] to study the effects of the door width, steps between the platform and the vehicle, and the fare collection on the boarding and alighting time. The door width of the train has been studied using experiments by [21], in which the authors found that the wider the doors, the lower the boarding and alighting time. However, the boarding and alighting time is also affected by the accessibility conditions between the train door and the platform, among other elements inside the train [22].

Another experiment [23] reported that using steps increased the boarding and alighting time, especially for passengers with luggage. Moreover, the difference between the train and the platform has been studied to improve the accessibility at the PTI [24]. Similarly, for passengers with reduced mobility the difference between the train and the platform should be reduced to its minimum value [25]. To improve the accessibility at the PTI, platform humps could be used to raise only a part of the platform and obtain level access to the train [26].

Other crowd management measures have been tested to study their effects on boarding and alighting time at the PTI. For example, using platform edge doors improves the safety conditions at the PTI. These elements have little impact on the boarding and alighting time but change the behavior of passengers as they know where the doors are located and, therefore, stay at the sides of the train doors rather than in front of them [27,28]. Moreover, the interactions between passengers boarding and alighting are affected by the ratio between them (e.g., the number of passengers boarding divided by those who are alighting), and therefore, real-time crowding information is needed to generate an adequate distribution of passengers at the platform and inside the train [29,30].

Recent studies have shown the distribution of passengers and their interaction based on variables such as queue length at the door, number of people inside the carriage, travel distance, and comfort of passengers [31]. The space of passengers and the occupation inside the train are reported in [32,33], which are essential factors that affect the interaction of passengers boarding or alighting. The characteristics of passengers are also studied in [34] utilizing laboratory experiments. The authors analyzed the behavior of passengers considering the ratio between passengers boarding and those who are alighting the train carriage. The behavior "disembarking precedes embarking" had the most significant impact on the boarding and alighting time. Moreover, the crowd behavior and distribution of passengers was evaluated by using a model to identify different stages and risks when boarding and alighting the train, affecting the operation safety and crowd accident prevention [35]. In the same line of research, [36] reported through a simulation model that waiting areas have an impact on the boarding and alighting process, improving the station level of service. This behavior is in concordance with other studies [37–41].

Although there have been significant developments in performed research, there is still a knowledge gap in identifying the effects of other crowd management measures, such as seat layouts inside the train, on the interactions between passengers boarding and alighting, which was the main objective of this study.

## 3. Experimental Method

### 3.1. Experimental Setup

To study the interactions of passengers boarding and alighting, a mock-up represented the dimensions of the platform–train interface (PTI). The mock-up was representative of existing stations in the Valparaiso Metro. According to a recent study [42], platforms in metro stations in the cities of Valparaiso and Viña del Mar have an average width of 3.0 m, and the trains are 2.5 m wide. In addition, train doors have a width of 1.3 m, and the seats inside the train are located perpendicular to the train's movement (see Figure 2).

Figure 3 shows the mock-up, which consists of a train carriage (7.0 m long by 2.5 m wide) and its adjacent platform (5.0 m long by 3.0 m wide) in front of one double door of 1.3 m wide. The platform included a yellow safety line (24 cm wide) representing the platform edge (60 cm from the train door). Sixteen seats and four vertical green handrails were considered inside the train carriage. To represent the same layout of an existing urban service in the Valparaíso Metro, a circular green horizontal handrail was used at the entrance hall of the train carriage.

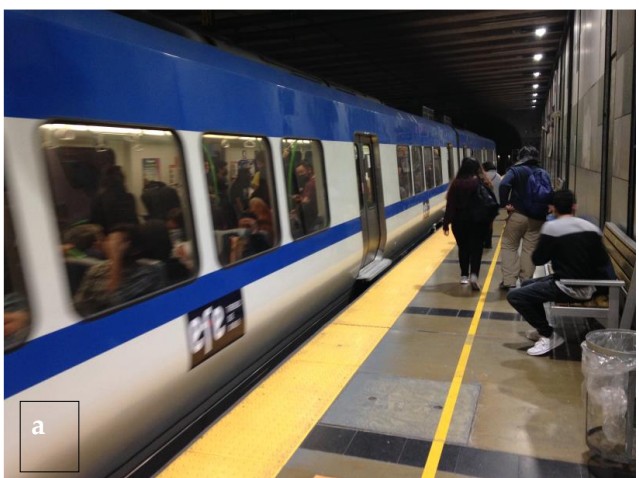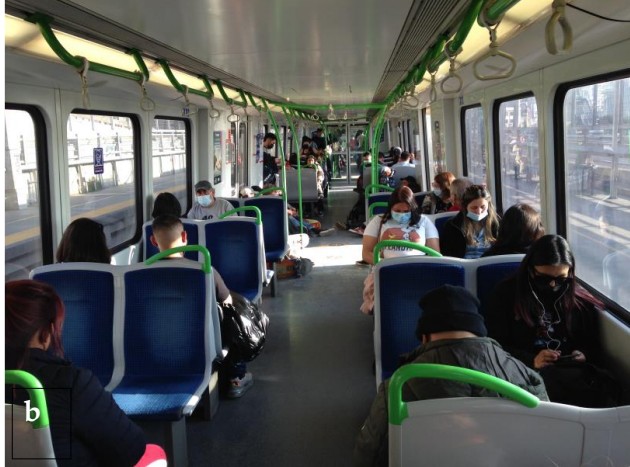

**Figure 2.** Layout of existing stations in the Valparaíso Metro: (**a**) platform and (**b**) inside the train.

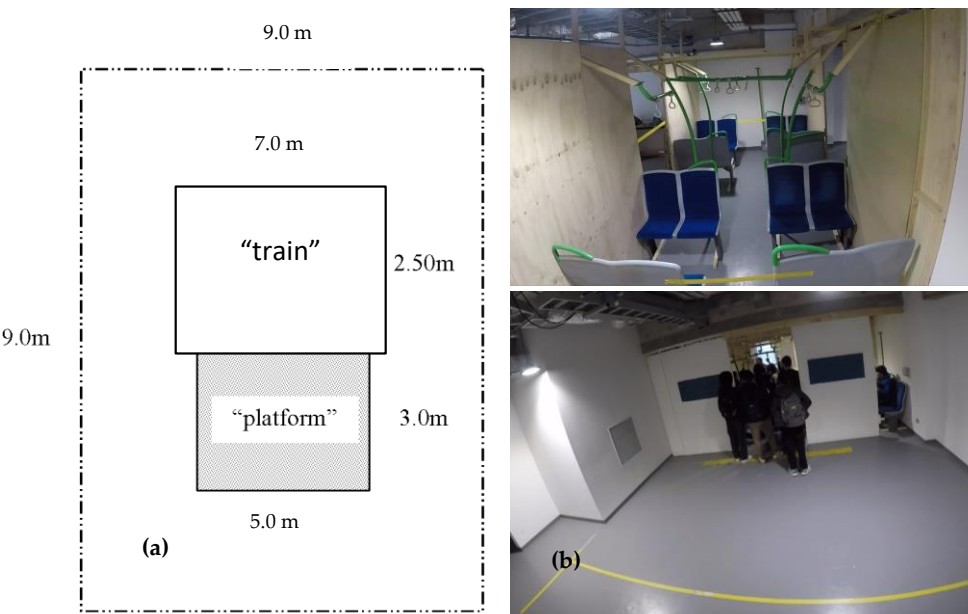

**Figure 3.** Mock-up in the controlled environment: (**a**) mock-up layout and (**b**) platform–train interface representing the train carriage and its adjacent platform.

The experiments considered 20 volunteers (50% male and 50% female) to represent the boarding and alighting of passengers at the PTI. The number of passengers was defined according to the density reported in [9,13], in which a value of 2 passengers/m² was reached. The volunteers were mainly students from university (80% of them were 18–25 years old, while the rest of them were 26–35 years old) who regularly used the urban services in the cities of Valparaíso and Viña del Mar (more than 65% of them used the system more than once a week). Almost half of the total number of volunteers used a backpack (9 volunteers) to represent a typical trip inside a metro station from or to the university. To identify their movements during the experiments, each volunteer was assigned a number from 1 to 20 fixed to their clothes.

In total, ten repetitions were performed per scenario, in which all volunteers boarded the train carriage from the platform. Once the doors were opened, all passengers alighted the train carriage. Therefore, three steps were considered in the experiments:

- Step 1: Passengers were waiting to board the train. The volunteers were distributed on the platform randomly.

- Step 2: The train doors were opened, and volunteers started to board the train carriage. Twelve seconds after the doors were opened, a sound was played to announce that the doors would be closed. This step was finished when the doors were closed. All volunteers boarded the train carriage and remained inside for about one minute to represent the journey between stations.
- Step 3: The train doors were opened, and volunteers started to alight the train carriage. Twelve seconds after the doors were opened, a sound was played to announce that the doors would be closed. This step was finished when the doors were closed. All volunteers alighted the train and remained on the platform for about 30 s to represent the time when the train was approaching the station. Then, they continued with step 1 as explained before.

It is essential to mention that a sound system was used when the train doors started to open. A similar sound was performed when the train doors started to close. These sounds were played through a microphone and a speaker system to replicate the same sound presented in existing stations.

In addition, to familiarize themselves with the experiments, volunteers did one repetition as a pilot test to recognize these three steps mentioned before. It is important to note that the sequence used in the experiments (including the pilot test) was based on previous studies performed in controlled environments [9,15,22,23], in which most participants were users from the metro service. Therefore, they represented a more realistic situation.

*3.2. Scenarios, Variables, and Detection Technique*

The experiments considered the following scenarios, in which the seat layout of the train carriage was changed. In total, ten repetitions per scenario were carried out, including a pilot test to recognize the process of boarding and alighting for volunteers. The scenarios are explained in the following (see Figure 4):

- Base scenario: The layout of the platform and the train carriage represented the current situation in existing stations. The platform was 3 m wide by 5 m long. In addition, the platform only considered a yellow safety line (20 cm wide) to represent the platform edge 60 cm from the train door. Therefore, the platform did not consider any other markings on the floor. Inside the train carriage, the layout was composed of 16 seats located perpendicular to the train's movement. In addition, four vertical green handrails and one circular horizontal green handrail were used inside the train carriage. The base scenario had a central hall of 3.25 m$^2$ (1.30 m by 2.50 m) and two corridors of 1.12 m$^2$ each (1.60 m by 0.70 m). The train area was equal to 11.25 m$^2$ (4.5 m by 2.5 m), and therefore, when all passengers were inside the train, a density of 1.77 passengers per square meter was obtained during the experiments (20 passengers divided by 11.25 m$^2$).
- Modified train scenario: The base scenario was modified to represent a different seat layout inside the train carriage, while the rest of the variables remained the same. As presented in Figure 4, this scenario considered the comparison between a train with seats in a perpendicular direction to the train movement and a train with seats in a parallel direction to the train movement. The modified train scenario had a central hall of 3.25 m$^2$ (1.30 m by 2.50 m) and two corridors of 2.41 m$^2$ (1.85 m by 1.30 m). The train area was equal to 12.5 m$^2$, and therefore, the train reached a maximum overall density of 1.60 passengers/m$^2$ (20 passengers divided by 12.5 m$^2$) during the experiments.

It is important to mention that volunteers were distributed inside the train randomly using the previous areas defined (i.e., they were told to board and alight the train as they were in an existing station, without assigning them to use a particular space inside the train).

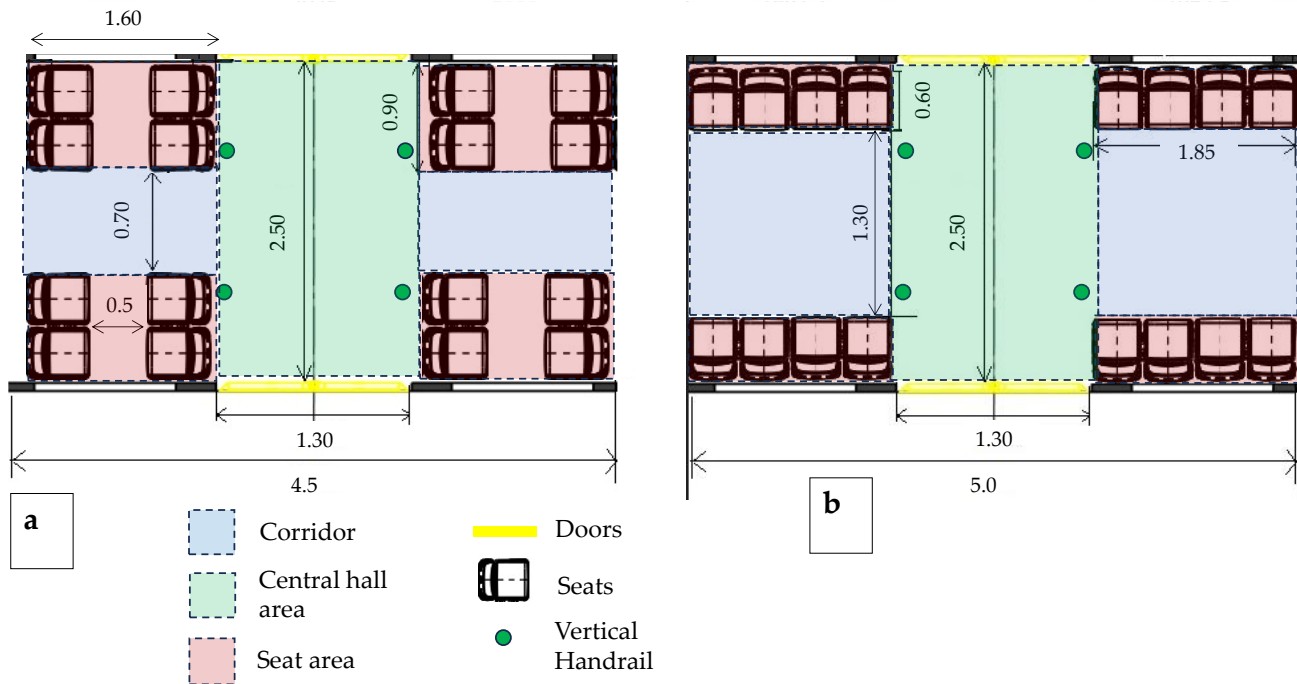

**Figure 4.** Layout of the train (dimensions in m): (**a**) base scenario (seats perpendicular to the movement of the train) and (**b**) modified train scenario (seats parallel to the movement of the train).

The following variables were defined to compare the scenarios. These variables were defined in concordance with a recent study conducted in existing stations in the Valparaiso Metro to study the behavior of passengers at the PTI [42]:

- Number of passengers at the central hall and corridors: This variable is defined as the number of passengers inside the train [passengers/m$^2$], divided into the entrance hall and corridors. The passenger number divided by the area of the entrance hall is defined as the density of the entrance hall. Similarly, the density in the corridors is reported as the passenger number divided by the area of the corridors. The density is related to the concept of level of service [10–12], which is used to describe other variables such as flow or speed of passengers.
- Boarding time and alighting time: The boarding time is defined as the difference in time between the first passenger who boarded the train and the last passenger who boarded the train [s]. Similarly, the alighting time is defined as the difference in time between the first passenger who alighted the train and the last passenger who alighted the train [s] [20,21,23,24].
- Occupation of seats: This variable is defined as the ratio between the number of seats occupied inside the train and the total number of seats inside the train [%]. For example, if all seats are occupied inside the train, this ratio equals 100% [30].

These variables were collected using a tracking tool based on a previous study [30]. The detection considered the counting of passengers and their trajectories using YOLO based on the algorithm reported in [43]. In Figure 5 a screenshot of this technique inside the train carriage is presented. Each volunteer was detected and labeled using a blue rectangle.

The results from the experiments were analyzed using a statistical technique. The Mann–Whitney U test was selected as it does not assume a distribution, considering independent observations. According to previous studies [9,15], this non-parametric test was used in pairwise comparison (e.g., two groups in experiment scenarios), using a small sample of data (e.g., 20 repetitions in the experiments).

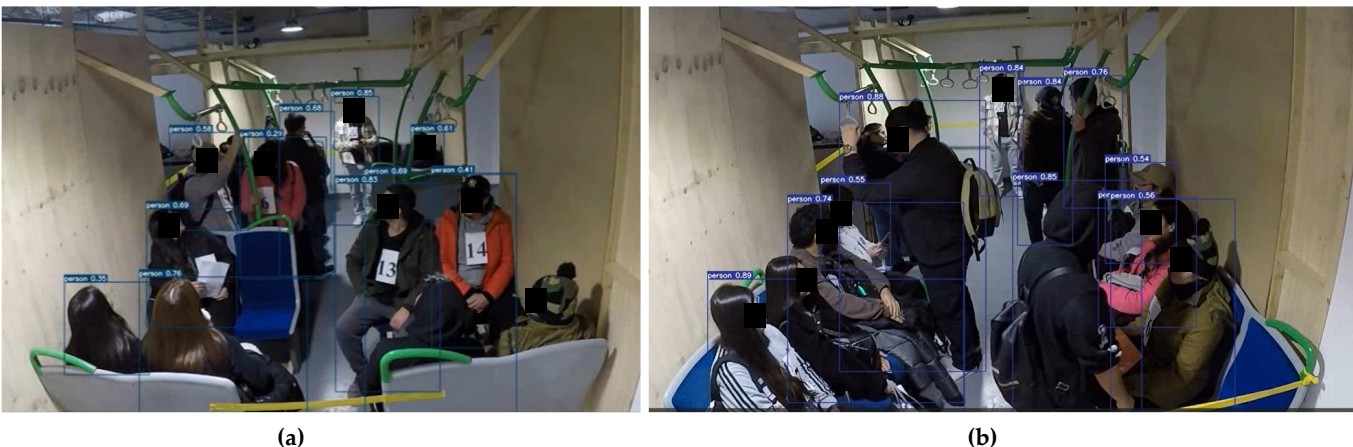

**Figure 5.** Example of detection technique screenshots: (**a**) base scenario (seats perpendicular to the movement of the train) and (**b**) train scenario (seats parallel to the movement of the train).

The null hypothesis (Ho) posited that the two medians were equal. To clarify, under the assumption of Ho, there existed no distinction in the cumulative sum of the two groups, meaning the sum of ranks for group 1 was equivalent to the sum of ranks for group 2. In instances where the combined sums of the two groups diverged, the outcomes suggested rejecting Ho at a significance level of $\alpha = 0.05$, implying statistical significance between the samples.

In addition, a questionnaire was completed after the experiments were performed. The questionnaire included some basic questions to classify participants (i.e., Gender? Age? How often do you use the metro system?) and other questions to obtain the perception of participants who represented the boarding and alighting process in each scenario on a scale from 1 to 5 or using "yes/no" questions (i.e., Did you consider adequate the capacity of the train carriage? Did you feel safe inside the train carriage? Did you recommend this configuration of seat layout for policy markers?).

## 4. Results

### 4.1. Number of Passengers in the Central Hall, Corridors, and Occupation of Seats

The experiments carried out were analyzed based on the number of passengers inside the train, including those located in the central hall after the boarding was finished. As shown in Tables 1 and 2, the average number of passengers in the central hall of the modified train scenario was reduced by 8% compared with the base scenario, reaching a value of 7.90 passengers on average ($\approx$8 passengers) and a standard deviation of 2.18 passengers ($\approx$2 passengers). This difference between scenarios represented only one passenger on average, and therefore, the Mann–Whitney U test ($\alpha = 0.05$) presented no statistical significance between the scenarios, i.e., the two medians were equal (the sum of ranks for the base scenario was no different from the sum of ranks for the modified train scenario). This could be because passengers in urban services prefer to stay closer to the train doors to be prepared for alighting while traveling short distances.

In the case of the corridors, the number of passengers in the modified train scenario increased three times compared with the base scenario, reaching a value of 2.90 passengers on average ($\approx$3 passengers) and a standard deviation of 0.88 passengers ($\approx$1 passenger) (see Tables 1 and 2). It is important to note that passengers were distributed randomly inside the train carriage, considering both corridors (left and right). The results showed that the null hypothesis was rejected, and therefore, there was statistical significance between the scenarios (i.e., the two medians were not equal). This could be caused by passengers preferring to reduce their interaction and stay in a space inside the train based on their comfort and safety, which is also in concordance with [9]. It is important to note that the modified train scenario registered an increase in the number of passengers in the corridors

due to the reduction in passengers at the central hall (one passenger on average), which was previously explained.

**Table 1.** Number of passengers inside the train after boarding was completed: base scenario.

| Number of Passengers | Repetition | | | | | | | | | | | |
|---|---|---|---|---|---|---|---|---|---|---|---|---|
| | 1 | 2 | 3 | 4 | 5 | 6 | 7 | 8 | 9 | 10 | Average | Std Dev. |
| Central hall | 5 | 4 | 11 | 6 | 9 | 9 | 9 | 10 | 11 | 12 | 8.60 | 2.72 |
| Corridors left ($C_L$) | 0 | 0 | 0 | 0 | 1 | 1 | 1 | 0 | 2 | 0 | - | - |
| Corridor right ($C_R$) | 0 | 1 | 0 | 0 | 1 | 0 | 0 | 0 | 2 | 1 | - | - |
| Total in corridors (($C_L$) + ($C_R$)) | 0 | 1 | 0 | 0 | 2 | 1 | 1 | 0 | 4 | 1 | 1.0 | 1.25 |
| Passengers seated on the left side of the train after boarding was finished ($S_L$) | 8 | 6 | 6 | 7 | 5 | 5 | 4 | 5 | 3 | 4 | 5.30 | 1.49 |
| Passengers seated on the right side of the train after boarding was finished ($S_R$) | 7 | 3 | 3 | 7 | 4 | 5 | 6 | 5 | 2 | 3 | 4.50 | 1.78 |
| The ratio between occupied seats and total seats inside the train carriage (($S_L$) + ($S_R$))/16 | 0.93 | 0.56 | 0.56 | 0.87 | 0.56 | 0.62 | 0.62 | 0.62 | 0.31 | 0.43 | 0.61 | 0.18 |

**Table 2.** Number of passengers inside the train after boarding was completed: modified train scenario.

| Number of Passengers | Repetition | | | | | | | | | | | |
|---|---|---|---|---|---|---|---|---|---|---|---|---|
| | 1 | 2 | 3 | 4 | 5 | 6 | 7 | 8 | 9 | 10 | Average | Std Dev. |
| Central hall | 3 | 11 | 7 | 9 | 7 | 8 | 7 | 9 | 8 | 10 | 7.90 | 2.18 |
| Corridors left ($C_{L\_M}$) | 2 | 0 | 2 | 1 | 1 | 1 | 1 | 2 | 2 | 2 | - | - |
| Corridor right ($C_{R\_M}$) | 1 | 4 | 0 | 1 | 1 | 3 | 2 | 0 | 1 | 2 | - | - |
| Total in corridors (($C_{L\_M}$) + ($C_{R\_M}$)) | 3 | 4 | 2 | 2 | 2 | 4 | 3 | 2 | 3 | 4 | 2.90 | 0.88 |
| Passengers seated on the left side of the train after boarding was finished ($S_{L\_M}$) | 7 | 4 | 4 | 5 | 6 | 4 | 6 | 4 | 6 | 4 | 4.78 | 0.97 |
| Passengers seated on the right side of the train after boarding was finished ($S_{R\_M}$) | 6 | 1 | 7 | 4 | 5 | 4 | 4 | 5 | 3 | 2 | 3.89 | 1.76 |
| Ratio between occupied seats and total seats inside the train carriage (($S_{L\_M}$) + ($S_{R\_M}$))/16 | 0.81 | 0.31 | 0.68 | 0.56 | 0.68 | 0.50 | 0.62 | 0.56 | 0.56 | 0.37 | 0.57 | 0.15 |

In addition, the difference between scenarios in the corridors was related to the number of seats occupied after the boarding of passengers was finished (e.g., passengers seated at the right or left of the train carriage). The results are presented in Tables 1 and 2, in which the ratio between the number of seats used and the total number of seats (16 in total) represents a value in the range of 0 to 1. For example, in the case of the base scenario, the first run reported a ratio of 0.93, which was calculated as the sum between the number of passengers who were seated at the left-hand side of the train carriage (eight passengers) and the number of passengers who were sitting on the right-hand side of the train carriage (seven passengers), divided by 16 (total number of seats inside the train carriage). On average, the modified train scenario presented a value equal to 0.57, which was 7% less than the base scenario (0.61), representing a reduction of one passenger on average. This could be interpreted as the case in which corridors were wider, and fewer passengers used the seats, which may be connected with a previous argument about the space availability and interaction between passengers. Even if there was a difference of one passenger between the scenarios, the Mann–Whitney U test ($\alpha = 0.05$) showed that the comparison between the samples did not present statistical significance. The null hypothesis was defined as the two medians being equal (i.e., the sum of ranks for the base scenario was no different from that of the ranks for the modified train scenario).

If the number of passengers was divided by the area of the train carriage, then the density was obtained. As presented in Figure 6, the modified train scenario reached a lower density at the central hall for most of the repetitions compared with the base scenario. In this case, the density was obtained by dividing the number of passengers at the central

hall by the area of the central hall. However, in the corridors, Figure 7 shows that the modified train scenario reached a higher density compared with the base scenario. This was because in the base scenario, few passengers were using the corridors of the train carriage due to reduced space available, affecting the interactions of passengers. In other words, in the modified train scenario, the use of seats parallel to the movement of the train generated a wider space in the corridors, and therefore, more passengers used this space compared with the base scenario. The density of passengers at the corridors was obtained by dividing the number of passengers at the corridors (left and right corridors) and the areas of those corridors.

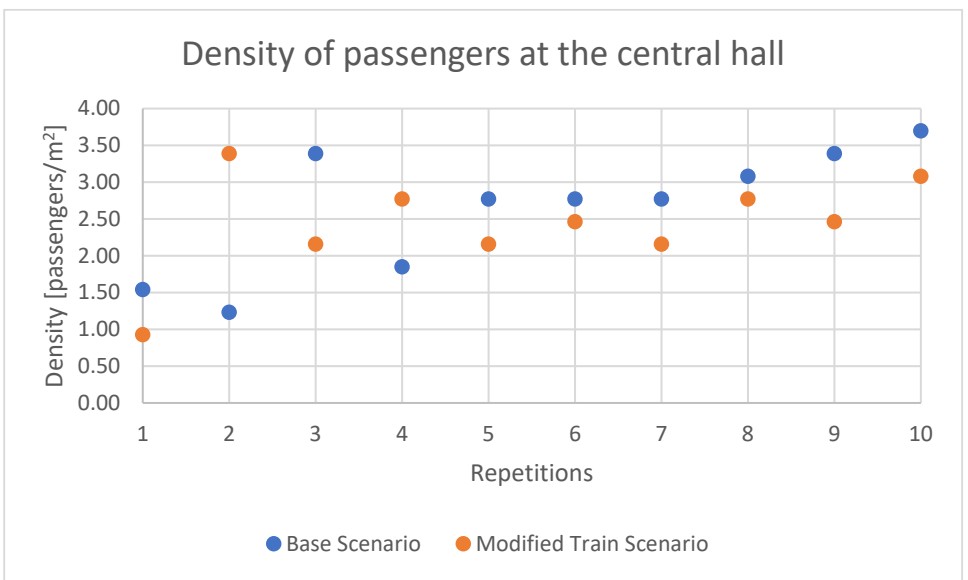

**Figure 6.** Density of passengers at the central hall of the train carriage: base scenario vs. modified train scenario.

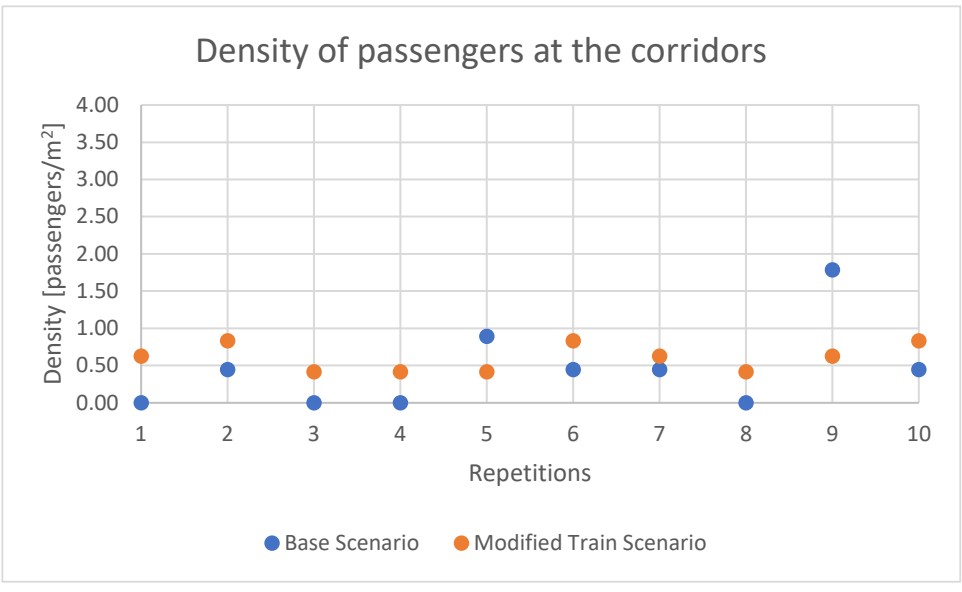

**Figure 7.** Density of passengers at the corridors inside the train carriage: base scenario vs. modified train scenario.

### 4.2. Boarding and Alighting Time

The boarding and alighting times were analyzed for each scenario (see Tables 3 and 4). The boarding time in the case of the base scenario presented an average of 13.80 s, while in the modified train scenario, a value of 13.09 s was reached, representing a reduction of

5%. In the case of the alighting time, the reduction was equal to 2% on average, reaching a value of 12.51 s in the case of the modified train scenario.

**Table 3.** Boarding time: base scenario vs. modified train scenario.

| Boarding Time | Repetition | | | | | | | | | | | |
|---|---|---|---|---|---|---|---|---|---|---|---|---|
| | 1 | 2 | 3 | 4 | 5 | 6 | 7 | 8 | 9 | 10 | Average | Std Dev. |
| Base scenario | 20.43 | 12.28 | 14.02 | 13.2 | 15.36 | 13.28 | 13.45 | 12.24 | 12.2 | 11.52 | 13.80 | 2.58 |
| Modified train scenario | 13.26 | 15.08 | 12.51 | 10.51 | 12.22 | 14.00 | 10.54 | 13.37 | 15.43 | 14.00 | 13.09 | 1.68 |

**Table 4.** Alighting time: base scenario vs. modified train scenario.

| Alighting Time | Repetition | | | | | | | | | | | |
|---|---|---|---|---|---|---|---|---|---|---|---|---|
| | 1 | 2 | 3 | 4 | 5 | 6 | 7 | 8 | 9 | 10 | Average | Std Dev. |
| Base scenario | 18.49 | 11.09 | 13.49 | 12.04 | 12.15 | 13.43 | 12.47 | 10.42 | 11.48 | 12.22 | 12.73 | 2.24 |
| Modified train scenario | 12.10 | 12.48 | 12.28 | 12.39 | 12.37 | 11.34 | 13.15 | 15.31 | 12.41 | 11.26 | 12.51 | 1.13 |

A Mann–Whitney U test ($\alpha = 0.05$) was performed to compare the scenarios. The null hypothesis was defined as the two medians being equal (i.e., the sum of ranks for the base scenario was no different from that of the ranks for the modified train scenario). However, in both cases, the scenarios did not present significant differences in the boarding and alighting times, which may be caused by a similar reason to that given previously, in which passengers preferred to stay closer to the train doors as they did not change their position even if they had more space in the corridors of the train carriage. This may affect the boarding and alighting time as the number of passengers at the central hall did not significantly differ between scenarios.

### 4.3. Perception of Comfort and Safety

The results obtained by the questionnaire are presented in Table 5, which shows that passengers perceived the modified train scenario as safer than the base scenario. In the case of the modified train scenario, 12 passengers perceived the seat layout as safer (seven passengers marked a scale of 4, and five passengers marked a scale of 5), which was 33% higher compared with the base scenario, in which nine passengers perceived the seat layout as safer (eight passengers marked a scale of 4, and one passenger marked a scale of 5). This variation came from those passengers who felt neutral in the base scenario and moved to a safer scale in the case of the modified train scenario. It is also interesting to note that in both scenarios, the number of passengers who felt unsafe was equal to two. It seemed that there was no difference for those two passengers when changing the seat layout, which could be caused by other factors such as density and location of vertical handrails, among others, which is concordance with previous studies [9,15,42].

**Table 5.** Perception of safety in each scenario after the experiments were conducted.

| Number of Participants Who Answered the Questionnaire | Scale for Safety Perception (1 = Very Low, 2 = Low, 3 = Neutral, 4 = High, 5 = Very High) | | | | |
|---|---|---|---|---|---|
| | 1 | 2 | 3 | 4 | 5 |
| Base scenario | 0 | 2 | 9 | 8 | 1 |
| Modified train scenario | 0 | 2 | 6 | 7 | 5 |

Concerning comfort, passengers were asked to express themselves about their perception of the capacity inside the train carriage (see Table 6). In the base scenario, only one passenger perceived that the capacity of the train carriage was adequate, while in the modified train scenario, nine passengers had the same perception (i.e., this perception increased

nine times). This increase was generated by passengers who changed their perception from not adequate to adequate. It is essential to highlight that still, four passengers did not change their perception between the scenarios. This could be caused by a similar reason to that presented previously, in which other factors affected the passengers´ perception.

**Table 6.** Perception of comfort in each scenario after the experiments were conducted.

| Number of Participants Who Answered the Questionnaire | Do you Consider Adequate the Capacity of the Train Carriage? | | |
|---|---|---|---|
| | Yes | No | I Do Not Know |
| Base scenario | 1 | 15 | 4 |
| Modified train scenario | 9 | 7 | 4 |

Finally, seven passengers recommended the base scenario when asked if they would recommend the seat layout for policymakers. However, in the case of the modified train scenario, 13 passengers recommended this configuration. In other words, most passengers preferred to have a train carriage in which seats were located parallel to the movement of the train rather than perpendicular to the train movement.

**5. Discussion and Conclusions**

This study focused on the interaction of passengers boarding and alighting when changing the seat layout inside the train carriage. An experimental approach was used based on laboratory experiments, in which 20 volunteers were recruited to represent the boarding and alighting processes.

The results of the experiments were interesting. Seats located parallel to the train movement (modified train scenario) reduced the interaction between passengers as it generated two corridors of 1.3 m width inside the train carriage. If the corridors were wider, then a greater space was available for passengers (the area of each corridor increased from 1.12 m$^2$ to 2.41 m$^2$). Therefore, the number of passengers staying in this space increased up to three times after the boarding was finished, presenting a statistical significance compared with the case in which the seats were located perpendicular to the train (base scenario). This could be because passengers may perceive the available space inside the train according to the comfort and safety conditions of the layout to reduce the interaction between them. This phenomenon concurs with the behavior reported in the literature, especially when studying the platform–train interface [9].

Concerning the central hall, the scenarios presented no statistical significance, which could be caused by the fact that passengers in urban services are located closer to the train doors as they are alighting in a station closer to the station where they board the train. In other words, passengers in urban services use the metro to move a few stations to their destination. Therefore, they prefer to stay in the central hall even if they have more space to move through the corridors of the train carriage, which is concordance with [30].

Regarding the boarding and alighting time, the scenarios did not present significant differences, which may be caused by a reason similar to that given before, in which passengers in urban services stay closer to the train doors even if they have more space to move in a direction to the corridors of the carriage. In concordance with a previous study [9] to reduce the boarding and alighting time, it is necessary to increase the number of flow lanes and, therefore, alight or board in the fastest way while also considering comfort and safety issues.

Finally, it is important to mention that the occupation of seats was independent of the layout inside the train. Even if passengers had more space to move in the corridor of the train carriage, they did not use more seats. More passengers preferred to stand in the corridor than to be seated. Although this may have a reason, passengers tried to reduce their interaction with other passengers as they may prefer to not sit closer to another passenger. Still, passengers used the seats. Therefore, even if trains have a new seat layout configuration, most passengers will still try to accommodate them by using the seats to

obtain a comfortable and safe space inside the train, which is an additional factor that could be complemented by previous studies [30].

Concerning the perception of safety and comfort, it seemed that passengers preferred to have a seat layout configuration parallel to the movement of the train, rather than perpendicular to the train movement. This perception was obtained from a questionnaire at the end of the experiments for each scenario. Consequently, for policymakers it is relevant to expand the experiments used in this paper to analyze new layout configurations inside the train (e.g., identify which layout is adequate when buying new rolling stock).

Further research will expand the possibility of including passengers remaining inside the train carriage while other passengers are alighting. In addition, passengers with reduced mobility will be considered to expand a previous study in which the space used by passengers may affect their interaction. Different density scenarios will be considered in future experiments (e.g., fewer or more than two passengers/m$^2$), in which the behavior of participants will be analyzed (e.g., including questions along the lines of "Did you behave as you normally did?" or "Why did you choose to stand/sit?" or "Why did you choose your position in the carriage?").

**Author Contributions:** Conceptualization, S.S., V.A. and T.F.; methodology, S.S. and A.L.; software, S.G. and G.B.; validation, S.G., G.B. and S.S.; formal analysis, S.G., G.B. and S.S.; investigation, S.S., V.A., A.L. and T.F.; resources, S.S.; data curation, S.S. and A.L.; writing—original draft preparation, S.S.; writing—review and editing, V.A., A.L. and T.F.; visualization, S.G., G.B. and S.S.; supervision, S.S.; project administration, S.S.; funding acquisition, S.S. All authors have read and agreed to the published version of the manuscript.

**Funding:** This research was funded by ANID, Chile grant FONDECYT Project 11200012 and FONDEF id22i10018. The APC was funded by ANID, Chile grant number FONDEF id22i10018.

**Institutional Review Board Statement:** The study was conducted in accordance with the Declaration of Helsinki, and approved by the Institutional Review Board (or Ethics Committee) of Universidad de Los Andes (protocol code CEC202089 approved date 23 October 2020) and Pontificia Universidad Católica de Valparaíso (protocol code BIOEPUCV-H 548-2022 approved date 3 October 022).

**Informed Consent Statement:** Informed consent was obtained from all subjects involved in the study.

**Data Availability Statement:** The data presented in this study are available on request from the corresponding author.

**Acknowledgments:** The authors would like to thank the volunteers who participated in the experiments. In particular, the authors are thankful for the collaboration between researchers from University College London, Universidad de Chile, and Pontificia Universidad Católica de Valparaíso, who shared some techniques and methods of study.

**Conflicts of Interest:** The authors declare no conflict of interest.

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
