# Peer review of "The Effect of Seat Layout on the Interaction of Passengers Inside the Train Carriage: An Experimental Approach for Urban Services"

_sustainability, doi:10.3390/su16030998_

Round 1
Reviewer 1 Report
Comments and Suggestions for Authors
The authors present an interesting study, using an interesting approach and present some nice findings. In general the manuscript provides an interesting read, using a fitting method which allows to study the effect of the seat layout. Such information can indeed be beneficial both from a scientific and practical standpoint. This being said, there are some issues with the reporting, making the manuscript feel a bit “rushed” since the authors skip over some important steps and explanations.
General comments
- Some minor language issues can be found, I suggest using Grammarly to go through the manuscript. Some examples are:
Row 22 “more closer” should be “closer”.
Row 71 “the laboratory setting” should be “a laboratory setting”.
Row 169 “using backpack” should be “using a backpack”
Row 255 “these variables were analyzed using a tracking tool” I assume this should be “these variables were collected using a tracking tool” since the tracking tool cannot analyze.
Although all minor, it does take away from the manuscript.
Section 2
The information presented in this section seems relevant, but not within the context of “interactions in the PTI” as the heading suggests
Row 92-99 for example feels out of place at the moment, as it dives into safety and some suggestions, whereas the heading talks about interactions. A similar note can be made about rows 101 to 108.
I believe this is not a matter of the information being presented but the context in which the authors present it, given the heading used for section 2. I would, therefore, suggest a different heading, maybe consider rephrasing as “literature review”.
Experimental set up
- The order of information when describing the experimental set up is off. When reading the description of the door width I was left wondering why these measurements were chosen, which was later described in row 173, this information should be provided earlier to clarify the choices made for the mock-up.
- In line with this, Figure 2b shows yellow markings using arrows, are these always present in the real-world scenario or is this part of the experiment. It is good to mention this as it made me wonder if it has an effect on the experiment presented in the manuscript or not.
- I am wondering if 20 volunteers is sufficient, I do not know how many one would need but it feels on the low side given how busy metros can get. Could you discuss on this, what the implications could be and if 20 participants is deemed to be sufficient for the purpose of the study?
- In line with this, participants were given 1 repetition as a test, is this enough, did you observe that it was clear to the participants what their task was and if they behaved in a normal manner?
- Row 181 mentions that a total of 20 repetitions were done, maybe mention that this was 10 per condition
- The “number of passengers” is defined as a variable, however it is stated in row 181 that all participants boarded and alighted the carriage. To me this sounds like the variables “number of passengers” should always be 20.
Results
- It was unclear to me what the areas mentioned in 268 meant, some labels could be added in Figure 4 to clarify this for the reader.
- The tables are unclear to me, it was mentioned that all participants boarded and alighted the carriage, but counting the passengers per repetition I do not count 20. For repetition 1 in Table 1 you only mention 5 passengers, where did the other 15 go?
- Table 5 shows the “Ratio between occupied seats and total seats inside the train carriage” with the following in brackets ((5)+(6))/16). Why is this equation in brackets added? It is unclear to me if this is an example of how it is calculated, or if it serves a different purpose. If it is there as an example I would suggest moving it into the body of the text instead.
- Why do Figures 6 & 7 start at 0 and not at 1?
- Why was the Mann Whitney U test chosen?
- In line with this, if using a non-parametric test
- An explanation for the non-significant findings is provided in row 312, is this based on observed behavior during the experiment or based on an assumption? If it is the former it is worth to clarify that this was observed if not, then
Discussion
- Row 352 mentions that the proportion of passengers in the corridor was more than expected, but the expected proportion was never mentioned.
- I am curious how there can be a statistically significant change in the corridors but the change in the central hall is not statistically significant.
- I am missing some suggestions of what these results mean in practice, what kind of layout should operators choose when buying rolling stock?
- I am curious why there was no post-experiment questionnaire to better understand some of the choices made by the participants. The point raised in row 372 is interesting for example, but could have benefited from such a questionnaire to raise it from an assumption to an observation.
- The future studies sound interesting!
Comments on the Quality of English Language
Could use proofreading and the use of Grammarly
Author Response
Thank you very much for your comments, we tried to address all of them as best as possible (please see attached).

Reviewer 2 Report
Comments and Suggestions for Authors
This paper focuses on the platform train interface (PTI) and conducts experimental research on the complex pedestrian motion patterns in this space. This is indeed one of the important topics in the current popular research field of pedestrian movement in rail transit. But due to the following issues with the article, I have to reject this submission:
1. There are also some minor formatting issues with the article, such as the lack of unified decimal places for the experimental scene size in Figure 2a) and Figure 4.
2. In section 3.1, it was mentioned that in order to simulate a common trip in a metro station from or to the university, nearly half of the experimental personnel carried backpacks, but the number of backpacks carried and the weight of the backpacks were not correspondingly controlled, which is not conducive to controlling the uniformity and authenticity of the experiment and may have a certain impact on the experimental results;
3. In section 3.1, the experimental simulation set up pedestrians to stay in the carriage for 1 minute after getting on the car, and then begin the alighting experiment. Is this time setting consistent with the actual situation? Will too short a time affect the movement behavior of pedestrians in the carriage, such as causing pedestrians to wait in the central hall near the car door instead of choosing to sit.
4. In section 3.2, the total space area of the carriages in the basic and improved scenarios is different. Will this have an impact on the boarding and alighting time in the research results.
5. The article lacks analysis of experimental data, and the conclusions drawn require more data support.
Comments on the Quality of English LanguageThe English writing should be improved.
Author Response

(The authors gave the same response as above.)

Reviewer 3 Report
Comments and Suggestions for Authors
This research concentrated on examining how changes in the seating arrangement within a train carriage affect the interaction between passengers during boarding and alighting. An experimental approach was employed, utilizing laboratory experiments that involved 20 volunteers representing the boarding and alighting processes.
The authors demonstrate that When the space in each corridor increased from 1.12 m2 to 2.41 m2, more passengers opted to use them after boarding the train. However, the actual increase in the number of passengers using the corridors exceeded the expected proportion. This discrepancy may be attributed to the available space, aiming for a more comfortable and secure layout within the train to diminish passenger interaction—a trend reminiscent of observations in other studies focusing on platform behavior.
Regarding the central hall, the presented scenarios did not exhibit statistical significance. This lack of significance could be explained by the fact that passengers in urban services tend to position themselves closer to the train doors when alighting, especially when disembarking at a station near their boarding point.
This manuscript introduces a compelling research study, and on the whole, it exhibits clarity and an appreciable level of articulation in its written presentation. The content is interesting and effectively communicates the research findings, contributing to the overall quality and coherence of the paper.
Introduction. Within this section, a comprehensive elucidation is provided, distinctly outlining the context and motivation that underscore the current work. This delineation emphasizes the critical significance of carefully designing the spatial arrangement of seats within metro trains. The rationale for such attention is underscored by the discernible repercussions on passenger interaction dynamics, as well as the consequential impact on safety considerations, particularly in instances of heightened congestion on trains and platforms.
Section2. While the exposition on the current state of the art is adequately presented, there exists an opportunity for further enhancement within this section by incorporating a more in-depth exploration. Specifically, there is merit in expanding the discourse to encompass methodologies directed towards augmenting passenger comfort. This augmentation can be achieved through the optimization of metro traffic control, thereby delving into a more comprehensive analysis that encompasses not only the design of the PTI and seats but further strategies aimed at improving the overall passenger experience, see e.g., A Service-Oriented Metro Traffic Regulation Method for Improving Operation Performance, Railway disruption: A bi-level rescheduling algorithm, Demand-Oriented Rescheduling of Railway Traffic in Case of Delays, Event-Triggered Predictive Control for Automatic Train Regulation and Passenger Flow in Metro Rail Systems, A Learning Based Intelligent Train Regulation Method With Dynamic Prediction for the Metro Passenger Flow, Robust fuzzy predictive control for automatic train regulation in high-frequency metro lines.
Sections 3 and 4. The experimental method is explained clearly, however it is not evident the novelty and contribution of the work. In fact, apart from the metro of Valparaiso, various other types of configurations of seats and vertical/horizontal bars have been implemented and are present in other metro lines in other regions or countries. Then, it is unclear why the authors build a mockup to test only two different seats and bars configuration, when such configurations are already implemented and currently used world-wide. The considered number of passengers is not motivated (only 20 appears too low) and an analysis with an increasing number of passengers would be beneficial to analyze the influence of seats and bar configurations when the carriage becomes overcrowded.
Comments on the Quality of English Language
The clarity of the English language in the document is appreciable, with only limited instances of minor typographical errors.
Author Response

(The authors gave the same response as above.)

Reviewer 4 Report
Comments and Suggestions for Authors
This paper studies the effect of seat layout on the interaction of passengers inside the train carriage. These are my comments on the paper:
1) What other factors can be taken into consideration for this study, to make it more comprehensive?
2) Why was the setup of the modified train scenario chosen this way? If the arrangement of the seats is changed, are we expecting change in the results?
Author Response

(The authors gave the same response as above.)

Round 2
Reviewer 1 Report
Comments and Suggestions for Authors
The researchers present an interesting piece of work that can help guide future design strategies for carriage design. Although the work has been improved from the previous version, I still have some language issues and a few examples below (this is not an extensive list):
Row 336: “The Mann-Whitney U-Test was selected as it is not assumed a distribution” should be “as it does not assume a distribution”
Row 339/340: ”The null hypothesis (Ho) was defined as the two medians were equal.” Should be “as the two medians being equal”
Row 368/420: “This could be caused because” should be “This could be caused because” OR “This could be caused by”
A thorough proofread would be beneficial for the quality of the work presented here.
The addition of the questionnaire results helps to improve the quality of the work, but for the future I would suggest to include/present questions regarding the behavior of participants. This could include questions along the lines of “did you behave as you normally did?” or “why did you choose to stand/sit?” or “why did you choose your position in the carriage”. This could help to lift some of the assumptions made, such made in row 368 and 420.
The authors mention “As a consequence, policymakers would con-659 sider this study for practical issues (e.g. when buying new rolling stock).” in row 659. This is quite a hasty statement at the moment, as you provide no clear clarification as to how and why. As I read it, this statement is only applicable to the subjective experience of passengers, but the different layout does not necessarily lead to operational benefits. Now, if this is indeed the case it is not a bad thing, but it is not clearly mentioned at the moment making the statement.
That being said, the work greatly improved and I would suggest minor revisions to fix the issues described above.
Comments on the Quality of English LanguageA thorough proofread would be beneficial for the quality of the work presented here.
Author Response
Thank you very much for your comments. Please see the document attached.

Reviewer 2 Report
Comments and Suggestions for Authors
--My comment 1: There are still some non-unified decimal places in Fig.4.
--All the format of figures and tables are hard for readers to see. There are many mistakes. Please correct all figures and tables in the whole paper in a formal and reasonable format.
--If you revise the paper, please mark the revised places in color, rather than using the revision mode. This will increase the disorder state of the paper.
--The contributions and aims are not reported in the Introduction.
--English writing still needs to be improved.
--Literature Review: The review lacks some related references in recent two years, such as:
Y. Zhou, J. Chen, M. Zhong, Z. Li, W. Zhou, Z. Zhou, Risk analysis of crowd gathering on metro platforms during large passenger flow, Tunnelling and Underground Space Technology, 142 (2023) 105421.
L. Fu, Q. Chen, Q. Shi, Y. Chen, Y. Shi, Characteristics of pedestrians’ alighting and boarding process in metro stations, Tunnelling and Underground Space Technology, 141 (2023) 105362.
Comments on the Quality of English Language
English writing still needs to be improved.
Author Response
Thank you very much for the comments. Please find atached the document with the responses.

Round 3
Reviewer 2 Report
Comments and Suggestions for Authors
It can be accepted.
Comments on the Quality of English LanguageIt is ok.
Author Response
Thank you very much.